# Common variants in *MMP20* at 11q22.2 predispose to 11q deletion and neuroblastoma risk

Xiao Chang[1], Yan Zhao[1], Cuiping Hou[1], Joseph Glessner[1], Lee McDaniel[2], Maura A. Diamond[2], Kelly Thomas [1], Jin Li[1], Zhi Wei[3], Yichuan Liu[1], Yiran Guo [1], Frank D. Mentch[1], Haijun Qiu[1], Cecilia Kim[1], Perry Evans[2], Zalman Vaksman[2], Sharon J. Diskin[2,4], Edward F. Attiyeh[2], Patrick Sleiman[1,4,5], John M. Maris[2,4] & Hakon Hakonarson[1,4,5]

*MYCN* amplification and 11q deletion are two inversely correlated prognostic factors of poor outcome in neuroblastoma. Here we identify common variants at 11q22.2 within *MMP20* that associate with neuroblastoma cases harboring 11q deletion (rs10895322), using GWAS in 113 European-American cases and 5109 ancestry-matched controls. The association is replicated in 44 independent cases and 1902 controls. Our study yields novel insights into the genetic underpinnings of neuroblastoma, demonstrating that the inherited common variants reported contribute to the origin of intra-tumor genetic heterogeneity in neuroblastoma.

[1] The Center for Applied Genomics, Children's Hospital of Philadelphia, Philadelphia, PA 19104, USA. [2] Division of Oncology and Center for Childhood Cancer Research, The Children's Hospital of Philadelphia, Philadelphia, PA 19104, USA. [3] Department of Computer Science, New Jersey Institute of Technology, Newark, NJ 07102, USA. [4] Department of Pediatrics, The Perelman School of Medicine, University of Pennsylvania, Philadelphia, PA 19104, USA. [5] Division of Human Genetics, Children's Hospital of Philadelphia, Philadelphia, PA 19104, USA. Correspondence and requests for materials should be addressed to J.M.M. (email: maris@email.chop.edu) or to H.H. (email: hakonarson@email.chop.edu)

Neuroblastoma, a malignant tumor of the developing sympathetic nervous system, remains among the most lethal and difficult to cure cancers that affect children[1]. Our previous GWA studies have revealed that common SNPs within or near *CASC15*, *BARD1*, *LMO1*, *HSD17B12*, *HACE1*, and *LIN28B* are associated with susceptibility to develop neuroblastoma[2–6]. The majority of neuroblastomas behave in highly malignant fashion, with widespread metastatic dissemination typical at the time of diagnosis. These "high-risk" neuroblastoma are characterized by genomic instability resulting in segmental chromosomal alterations and high-level amplification of the *MYCN* gene locus. These are prognostically relevant, with *MYCN* amplification (MNA), 11q deletion and 1p deletion currently used as biomarkers to assign therapy, but the mechanisms by which these chromosomal rearrangements arise remain obscure. MNA and 11q deletion have been shown to be almost mutually exclusive[7], implying two distinct subtypes underlying high-risk neuroblastoma. As the genetic contribution to neuroblastoma susceptibility is highly heterogeneous and complex, additional loci may be associated with a specific type of chromosomal abnormality in neuroblastoma.

Here we conducted GWASs for three subtypes of neuroblastoma, including MNA, 11q-deletion and 1p-deletion neuroblastoma. We identify a susceptibility locus in *MMP20* for 11q-deletion neuroblastoma and subsequently replicate interesting markers in an independent cohort. Our study provides new insights into the genetic architecture of neuroblastoma subtypes and suggests directions for subset-specific therapeutic strategies.

## Results

**Identification of prognostic biomarkers**. To profile genomic alterations of neuroblastoma, we genotyped 641 neuroblastoma tumor DNA samples, using Illumina HumanHap550 or 610 SNP arrays including 442 samples with matched blood DNA samples (Supplementary Data 1). Tumor-based copy number segments were first calculated by ASCAT and OncoSNP, and then curated manually. Tumor samples were excluded if the copy number results were suggestive of noise or if no amplification/deletion events were detected. Following thorough quality control, 437 tumor samples were kept for analysis, including 300 with matched blood DNA genotyping data. A total of 154 neuroblastoma tumors were identified with MNA, 245 with 11q deletion, and 167 with 1p deletion, respectively. The inverse correlation between MNA and 11q deletion events was also confirmed (Pearson correlation coefficient = −0.83), only 27 neuroblastoma tumors were identified with both MNA and

11q deletion. Besides the commonly observed MNA, 11q deletion and 1q deletion variations in neuroblastoma (Supplementary Fig. 1), we also identified a number of genomic regions targeted by low frequency somatic aberrations (Supplementary Figs. 2–7 and Supplementary Tables 1 and 2). In this regard, focal amplifications of *ALK*, *CCND1*, *LIN28B*, *MDM2*, and 19q13.42 observed in our study have been previously implicated in individual neuroblastoma studies (Supplementary Figs. 2 and 4)[8–12]. However, to our knowledge, recurrent focal amplifications of *MYC*, *ZFHX3*, *KRAS*, *RRAS2*, and *CYTH1* have not been reported in neuroblastoma primary tumors before (Supplementary Note 1 and Supplementary Fig. 3). Interestingly, we observed a number of cases (56/628) with low-level amplification of *CCND1* (8.9%). In most of those cases, the amplification of *CCND1* co-occurred with 11q deletion (53/56, 94.6%), suggesting these two events are highly correlated (Supplementary Note 1).

**Discovery stage**. We subsequently identified 113 11q-deletion cases of European-American ancestry with available blood DNA genotyping data, and we performed GWAS in this subset together with 5109 ancestry-matched controls (Supplementary Fig. 8). The genomic inflation factor was 1.0 (Supplementary Fig. 9). Three SNPs within *MMP20* at 11q22.2 (rs10895322, $P = 2.62 \times 10^{-9}$, OR = 2.858; rs3781788, $P = 2.46 \times 10^{-8}$, OR = 2.505; rs2280211, $P = 3.11 \times 10^{-9}$, OR = 2.604, logistic regression test) were found to surpass the conventional genome-wide significance threshold ($P = 5 \times 10^{-8}$, logistic regression test, Table 1 and Supplementary Fig. 10). All three SNPs map to the intronic regions of *MMP20* and showed a high degree of linkage disequilibrium (Supplementary Fig. 11). Considering that three case subgroups were investigated in this study, we used an adjusted threshold by dividing the conventional threshold by three ($P = 1.7 \times 10^{-8}$, logistic regression test) with the top SNP rs10895322 remaining significant. The reported SNPs at previously identified neuroblastoma susceptibility loci, including 2q35 (*BARD1*), 6p22.3 (*CASC15*), and 11p15.4 (*LMO1*) were also nominally associated with 11q deletion (Supplementary Table 3). As the 11q22.2 region harboring *MMP20* is commonly deleted in 11q-deletion neuroblastomas, we investigated 11q-deletion cases that are heterozygous (G/A) for rs10895322 and found that the risk allele (G) is preferentially retained in tumors ($P = 2.09 \times 10^{-3}$, binomial test). This result is consistent with the observation of preferential allelic imbalance in other cancers (Supplementary Fig. 12)[13, 14]. We did not observe evidence for epistasis between the previously reported loci and the newly discovered

**Table 1 Association results for the top three genotyped SNPs at 11q22.2**

| SNP | Study | A1/A2 | Freq1 | Freq2 | OR | 95% CI | P value |
|---|---|---|---|---|---|---|---|
| rs10895322 | Discovery | G/A | 0.164 | 0.064 | 2.858 | 1.991–4.101 | $2.62 \times 10^{-9}$ |
| | Replication | G/A | 0.131 | 0.057 | 2.478 | 1.296–4.740 | $4.58 \times 10^{-3}$ |
| | Combined | | | | 2.763 | 2.015–3.789 | $2.89 \times 10^{-10}$ |
| rs3781788 | Discovery | T/C | 0.201 | 0.091 | 2.505 | 1.794–3.497 | $2.46 \times 10^{-8}$ |
| | Replication | T/C | 0.148 | 0.087 | 1.813 | 0.995–3.301 | 0.0486 |
| | Combined | | | | 2.321 | 1.733–3.106 | $1.58 \times 10^{-8}$ |
| rs2280211 | Discovery | C/T | 0.208 | 0.092 | 2.604 | 1.875–3.615 | $3.11 \times 10^{-9}$ |
| | Replication | C/T | 0.148 | 0.088 | 1.795 | 0.986–3.269 | 0.0525 |
| | Combined | | | | 2.390 | 1.792–3.187 | $3.16 \times 10^{-9}$ |

A1/A2: risk allele/protective allele
Freq1: case frequency
Freq2: control frequency
P: P value calculated by logistic regression test

**Table 2 Results from subtype analysis of neuroblastoma risk loci**

| SNP | Overall logistic | | Subset search (case–control) | | | Subset search (case–complement) | | |
|---|---|---|---|---|---|---|---|---|
| | P value | OR (95% CI) | P value | OR (95% CI) | Best Subset | P value | OR (95% CI) | Best Subset |
| 11q22.2 MMP20 | | | | | | | | |
| rs10895322 | 2.55E-03 | 1.5 (1.153–1.952) | 2.98E-08 | 2.828 (1.958–4.084) | 11q-del | 2.85E-08 | 2.83 (1.96–4.087) | 11q-del |
| rs3781788 | 1.46E-04 | 1.549 (1.236–1.942) | 8.27E-09 | 2.682 (1.918–3.752) | 11q-del | 9.88E-09 | 2.671 (1.909–3.738) | 11q-del |
| rs2280211 | 4.54E-04 | 1.506 (1.198–1.893) | 9.77E-09 | 2.673 (1.91–3.74) | 11q-del | 1.05E-08 | 2.667 (1.906–3.732) | 11q-del |
| 2q35 BARD1 | | | | | | | | |
| rs3768716 | 1.39E-12 | 1.795 (1.527–2.11) | 1.22E-12 | 1.986 (1.643–2.399) | 11q-del,MNA | 2.09E-12 | 1.795 (1.525–2.113) | 11q-del,MNA |
| rs17487792 | 1.10E-12 | 1.799 (1.53–2.114) | 7.58E-13 | 1.992 (1.65–2.406) | 11q-del,MNA | 1.61E-12 | 1.966 (1.63–2.372) | 11q-del,MNA |
| rs7587476 | 1.84E-12 | 1.769 (1.51–2.074) | 2.18E-12 | 1.949 (1.618–2.348) | 11q-del,MNA | 2.87E-12 | 1.769 (1.507–2.076) | 11q-del,MNA |
| 6p22.3 CASC15 | | | | | | | | |
| rs4712653 | 2.12E-11 | 1.719 (1.467–2.015) | 7.42E-11 | 1.719 (1.461–2.024) | 11q-del,MNA | 7.45E-11 | 1.719 (1.461–2.024) | 11q-del,MNA |
| rs9295536 | 5.07E-11 | 1.691 (1.446–1.979) | 9.90E-11 | 1.691 (1.442–1.984) | 11q-del,MNA | 9.94E-11 | 1.691 (1.442–1.984) | 11q-del,MNA |
| rs6939340 | 5.58E-11 | 1.698 (1.45–1.99) | 1.36E-10 | 1.698 (1.445–1.996) | 11q-del,MNA | 1.37E-10 | 1.698 (1.445–1.996) | 11q-del,MNA |
| 11p15.4 LMO1 | | | | | | | | |
| rs110419 | 2.28E-03 | 1.27 (1.089–1.481) | 2.44E-03 | 1.605 (1.182–2.179) | 11q-del | 2.93E-03 | 1.593 (1.172–2.166) | 11q-del |

Association results are reported from the analysis of the newly discovered *MMP20* locus, and three previously reported loci including *BARD1*, *CASC15*, and *LMO1*. P values are calculated by ASSET analysis

**Table 3 Imputed SNPs surpassing genome-wide significance within and near MMP20**

| SNP | POS | A1 | Discovery | | Replication | | Combined | |
|---|---|---|---|---|---|---|---|---|
| | | | OR | P value | OR | P value | OR | P value |
| rs11225333 | 102454685 | A | 2.838 | $4.34 \times 10^{-7}$ | 2.206 | 0.012 | 2.673 | $9.71 \times 10^{-10}$ |
| rs5024119 | 102463359 | G | 2.861 | $4.35 \times 10^{-7}$ | 2.370 | 0.007 | 2.736 | $4.07 \times 10^{-10}$ |
| rs10895322 | 102470256 | A | 2.854 | $4.64 \times 10^{-7}$ | 2.361 | 0.007 | 2.728 | $4.54 \times 10^{-10}$ |
| rs3781788 | 102477556 | C | 2.616 | $1.92 \times 10^{-7}$ | 1.813 | 0.048 | 2.404 | $2.42 \times 10^{-9}$ |
| rs7115479 | 102483150 | C | 2.617 | $1.88 \times 10^{-7}$ | 1.809 | 0.049 | 2.403 | $2.44 \times 10^{-9}$ |
| rs7122793 | 102484945 | G | 2.559 | $3.60 \times 10^{-7}$ | 1.657 | 0.103 | 2.322 | $2.79 \times 10^{-8}$ |
| rs7123742 | 102484946 | C | 2.629 | $1.40 \times 10^{-7}$ | 1.650 | 0.106 | 2.372 | $9.75 \times 10^{-9}$ |
| rs7126560 | 102485553 | G | 2.636 | $1.53 \times 10^{-7}$ | 1.810 | 0.049 | 2.417 | $1.93 \times 10^{-9}$ |
| rs17099063 | 102487065 | C | 2.610 | $2.02 \times 10^{-7}$ | 1.797 | 0.052 | 2.395 | $2.82 \times 10^{-9}$ |
| rs2280211 | 102488131 | A | 2.604 | $2.14 \times 10^{-7}$ | 1.795 | 0.052 | 2.390 | $3.09 \times 10^{-9}$ |
| rs11225344 | 102493269 | G | 2.625 | $2.04 \times 10^{-7}$ | 1.931 | 0.024 | 2.443 | $2.95 \times 10^{-9}$ |
| rs2292731 | 102496405 | C | 2.425 | $2.08 \times 10^{-7}$ | 1.637 | 0.075 | 2.211 | $1.93 \times 10^{-8}$ |
| rs12786739 | 102499728 | T | 2.417 | $2.09 \times 10^{-7}$ | 1.631 | 0.077 | 2.204 | $2.15 \times 10^{-8}$ |
| rs12798540 | 102508004 | G | 2.866 | $4.59 \times 10^{-8}$ | 1.385 | 0.358 | 2.492 | $1.31 \times 10^{-8}$ |
| rs12575154 | 102534875 | C | 2.792 | $1.81 \times 10^{-7}$ | 1.470 | 0.296 | 2.473 | $3.14 \times 10^{-8}$ |
| rs34902925 | 102550821 | A | 2.796 | $1.93 \times 10^{-7}$ | 1.484 | 0.285 | 2.480 | $2.92 \times 10^{-8}$ |

POS: genomic position in human genome build hg19
A1: coded allele
P: P value calculated by frequentist association test

*MMP20* locus (Supplementary Tables 4 and 5), suggesting that these susceptibility loci contributed to disease risk, independently.

We also performed SNP association analysis in cases with MNA and 1p-deletion, respectively. The previously reported loci on 2q35 (*BARD1*) and 6p22.3 (*CASC15*)[2, 3], reached genome-wide significance (Supplementary Fig. 3 and Supplementary Table 3) in the GWAS of 260 MNA cases (Methods) and 5109 controls and the 6p22.3 locus also showed a P value near genome-wide significance in the GWAS of 69 1p-deletion cases and 5109 controls. However, no significant associations at 11q22.2 was detected in either MNA or 1p-deletion neuroblastoma (Supplementary Table 3). When GWAS was performed in the 113 11q-deletion cases and 282 controls (78 undeleted 11q and 204 MNA neuroblastomas, Methods) of European-American ancestry, the 11q22.2 locus was still nominally significant (rs10895322, $P = 5.50 \times 10^{-5}$,

OR = 2.811, logistic regression test, Supplementary Table 6), indicating a unique and independent role of the 11q22.2 locus in the 11q deletion cases. We subsequently applied the subset-based approach ASSET to investigate the impact of the 11q22.2 (*MMP20*), 2q35 (*BARD1*), 6p22.3 (*CASC15*), and 11p15.4 (*LMO1*) loci on the subtypes. ASSET is designed for a single case–control study in which cases are treated as distinct disease subtypes[15]. This permits both case–control and case–case comparisons (among subsets of disease subtypes) for the detection of the strongest association signals. Here, 11q deletion and MNA subtypes were included for the ASSET analysis, as they are negatively correlated and represent two distinct subtypes of neuroblastoma. However, 1p deletion was excluded as an independent subtype, since 1p deletion often co-occurs with 11q deletion or MNA. The ASSET results confirm the association between 11q22.2 and 11q-deletion subtype (Table 2).

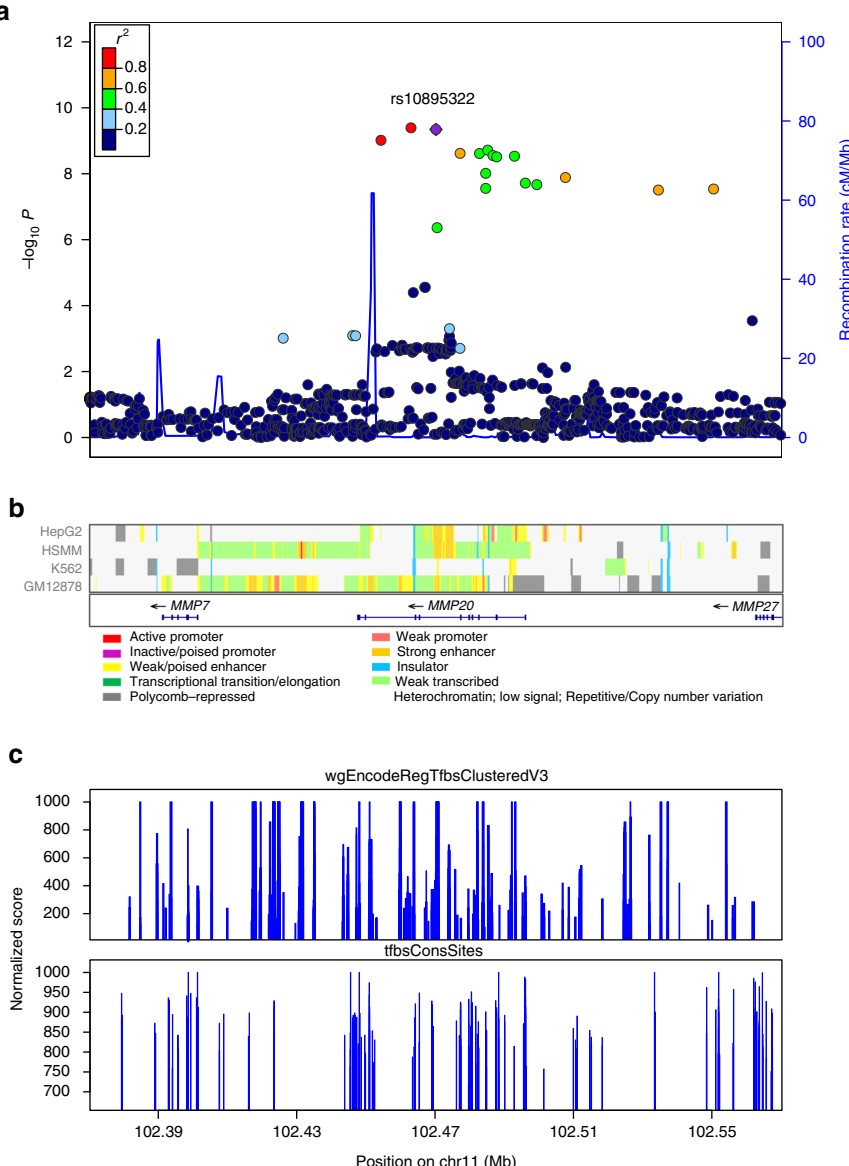

**Fig. 1** Regional association plot including both genotyped and imputed SNPs at 11q22.2. **a** Plotted are the regional association results from the meta-analysis of discovery and replication cohorts (−log10 transformed $P$ values) and the recombination rate. SNPs are colored to reflect pairwise LD ($r^2$) with the most significantly associated SNP. The most significant SNP is shown in *purple*. The tracks of **b** CHROMHMM (Celline: HepG2, HSMM, K562, GM12878), **c** wgEncodeRegTfbsClusteredV3 and tfbsConsSites annotation are plotted on the bottom

**Replication stage**. To replicate our findings, we genotyped the tumor and blood DNA samples of additional 192 neuroblastoma cases using OMNI-Express SNP array. Copy number analysis of the tumor samples identified 80 of them with 11q deletion, Among the 80 11q-deleltion cases, we identified 44 cases of European-American ancestry. The 44 cases and 1902 ancestry-matched controls were included for GWAS replication (Supplementary Fig. 8). All of the three genome-wide significant SNPs (rs10895322, $P = 2.62 \times 10^{-9}$, OR = 2.858; rs3781788, $P = 2.46 \times 10^{-8}$, OR = 2.505; rs2280211, $P = 3.11 \times 10^{-9}$, OR = 2.604, logistic regression test) at 11q22.2 in the discovery cohort showed the same direction of association in the replication set, and two of them had a $P$ value <0.05 (Table 1). The association was further strengthened by a meta-analysis combing the discovery and replication studies (Table 1).

**Imputation analysis**. To evaluate additional variants not assayed directly on the genotyping arrays, we imputed unobserved genotypes at 11q22.2 using 1000 Genomes Project data as the reference. Imputation identified 13 additional SNPs that were significantly associated with neuroblastoma (Table 3). The top SNPs, both genotyped and imputed were all located in a strong linkage disequilibrium region ($r^2 > 0.8$) within *MMP20* (Fig. 1 and Supplementary Fig. 11).

**Exome sequencing analysis**. To investigate the possibility of association being a consequence of a rare variant, we analyzed exome sequencing data from 229 neuroblastoma samples and whole-genome sequencing data from 143 neuroblastoma samples from the TARGET database (https://ocg.cancer.gov/programs/target/). We examined for the presence of low

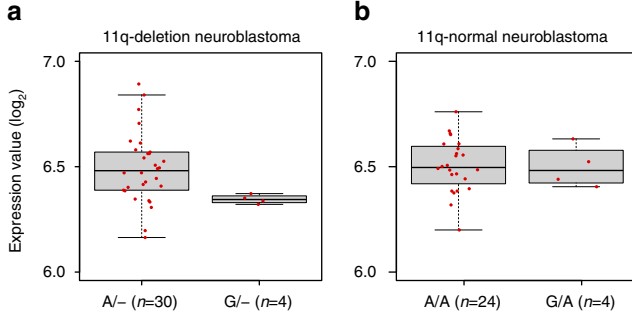

**Fig. 2** Box plot of *MMP20* mRNA expression levels in neuroblastoma samples with rs10895322 genotypes. **a** Plot of 34 11q-deletion neuroblastoma samples with rs10895322 genotype A/− ($n = 30$) or G/− ($n = 4$). **b** Plot of 28 11q-normal neuroblastoma samples with rs10895322 genotype A/A ($n = 24$) or G/A ($n = 4$). *Boxes* in boxplots represent first to third quartiles and *whiskers* extend to furthest data point still within 1.5 IQRs of either quartile. *Center lines* in boxes represent medians

frequency variants (<0.01 in 1000 Genomes Database) detected in the *MMP20* coding region. Only one pathogenic variant was detected, which is associated with amelogenesis imperfecta. In addition, the frequency of rare variants presents in neurobalstoma cases is not significantly different in the 1000 Genome Database, indicating that the association we report here is unlikely a consequence of a rare variant (Supplementary Data 2).

**eQTL analysis**. To gain further insight into the biological mechanisms of the association at 11q22.2, we performed an expression quantitative trait loci (eQTL) analysis using gene expression data from 34 11q-deletion neuroblastoma cases. eQTL analysis showed that rs10895322 is significantly associated with *MMP20* expression in samples with 11q deletion ($P = 7.7 \times 10^{-5}$, Student's *t*-test, Fig. 2). We did not observe association between rs10895322 genotypes and the expression level of other genes near *MMP20* in the 11q-deletion samples. In addition, no significant association was detected between rs10895322 and *MMP20* expression levels in undeleted 11q neuroblastoma samples ($P = 0.94$, Student's *t*-test, Fig. 2).

**Discussion**

*MMP20* belongs to Matrix metalloproteinase (MMP) gene family, which is involved in tooth enamel formation and defects of *MMP20* have been associated with amelogenesis imperfecta[16]. A recent GWAS reported association of *MMP20* with neovascular lesion size in age-related macular degeneration (AMD), suggesting a potential role for anti-vascular endothelial growth factors (anti-VEGFs) drugs in reversing the choroidal neovascularization (CNV) disease process[17]. Interestingly, the lead SNP identified in our GWAS of 11q-deletion neuroblastoma (rs10895322), is also the top SNP reported in the GWAS of neovascular lesion size in AMD[17]. The risk allele of rs10895322 in our study is also the same allele that promotes neovascular lesion size in the GWAS of AMD, implying that the risk attributed to rs10895322 in neuroblastoma may be mediated through comparable angiogenesis process that is regulated by *MMP20*. Our eQTL analysis further indicated that rs10895322 may regulate *MMP20* expression specifically in 11q-deletion neuroblastoma cases. Given that knockdown of *MYCN* expression has been shown to significantly block VEGF secretion in neuroblastoma cells with MNA, but not in neuroblastoma

cells without MNA, it is conceivable that distinct molecular mechanisms of angiogenesis may underlie the MNA and non-MNA neuroblastoma disease process[18].

In summary, here we demonstrate that common variants at the *MMP20* locus are exclusively associated with the 11q-deletion subtype of neuroblastoma, indicating that the inherited common variants may contribute to the origin of intra-tumor genetic heterogeneity in neuroblastoma. In addition, we have refined our understanding of somatic chromosomal structural rearrangements in the high-risk neuroblastoma genome. Future studies addressing the mechanism by which *MMP20* impacts tumorigenesis of the 11q-deleted subset of neuroblastoma, which may allow for the development of novel molecular subset-specific therapeutic strategies.

**Methods**

**Sample collection**. The neuroblastoma cases in the study were individuals diagnosed with neuroblastoma or ganglioneuroblastoma registered through the Children's Oncology Group (COG). The tumor and blood samples were obtained through the COG Neuroblastoma bio-repository for specimen collection at the time of diagnosis. The majority of specimens were annotated with clinical and genomic information containing age at diagnosis, site of origin, INSS disease stage[19], International Neuroblastoma Pathology Classification[20].

Control subjects were recruited by the Center for Applied Genomics through the Children's Hospital of Philadelphia (CHOP) Health Care Network, including four primary care clinics and several group practices and outpatient practices that included wellchild visits. Eligibility criteria for control subjects were (i) selfreporting as Caucasian and (ii) no serious underlying medical disorder, including cancer. The Research Ethics Board of CHOP approved the study. Written informed consent was obtained from all subjects by nursing and medical assistant study staff under the direction of CHOP clinicians.

**Calculation of somatic copy number aberrations**. Copy number segments of the tumor samples were predicted with ASCAT, which accounts for the normal cell contamination and tumor aneuploidy[21]. Previous study pointed out that the SNP genotyping arrays showed a "genomic wave patterns" issue in which signal intensity was correlated to local guanine-cytosine content. The genomic_wave.pl program in the PennCNV package was used to adjust the signal intensity value[22].

**Genome-wide association studies of three subtypes**. We identified and report on 300 neuroblastoma samples with matched blood/tumor genotyping data. Among them, a total of 125 1p-deletion, 171 11q-deletion, and 101 MNA cases were identified. After population stratification, 78 1p-deletion, 113 11q-deletion, and 56 MNA cases of European ancestry were included for the association analysis. In addition, we genotyped the blood DNA samples of 299 MNA cases (*MYCN* amplification was identified by fluorescence in situ hybridization (FISH) as previously described[23]). Hybridization studies were performed with a cosmid probe (from the *MYCN* genomic locus on chromosome 2) that had been nick-translated with digoxigenin-dUTP. The labeled probe was combined with human Cot1 DNA and allowed to hybridize overnight at 37℃ to fixed tumor cells. Specific hybridization signals were detected by incubating the hybridized slides in a solution containing fluorescein-conjugated antidigoxigenin antibodies. Probe detection for two-color experiments included Texas red avidin and counterstaining with 4′, 6-diamidino-2-phenylindole (DAPI). Fluorescence microscopy was performed with a Zeiss microscope equipped with either fluorescein filter sets or a three-color filter set for FITC, Texas red, and DAPI. Two-color FISH was performed with a biotin-labeled chromosome 2 centromere-specific probe. 204 of the 299 MNA cases (detected by FISH) are European decedents. Finally, 260 MNA cases of European ancestry were included. The same set of ancestry-matched controls (5109 individuals) was used for the 11q-deletion, 1p-deletion and MNA GWAS.

**Quality control**. SNPs were filtered by genotype missing rate (<0.95), minor allele frequency (<0.01), and Hardy–Weinberg equilibrium *P* value (<0.00001).

Samples with a genotype call rate below 95% were excluded. To remove cryptic relatedness between samples, the identity-by-descent (IBD) scores were calculated and one individual in the pairs of subjects was removed with IBD>0.25. Principal component analysis were performed by using EIGENSTRAT to detect and correct for potential substructures and outliers[24].

**Statistical analysis**. The association analysis were carried out in PLINK using logistic test. The genomic inflation factors for 11q deletion, 1-p deletion and MNA GWAS were 1. Pairwise epistasis tests between the top significant SNPs at 11q22. 2

were also conducted with PLINK. Meta-analysis was performed by GWAMA[25]. Fixed effects *P* values were reported.

Cases and controls were pruned to a common set of SNPs before imputation. Genotype imputation at the 11q22.32 locus was performed with IMPUTE2 using the reference panel 1000 Genome Phase I integrated variants set[26]. We used SHAPEIT recommended by Howie et al to infer the haplotypes before imputation[27]. In consideration of the uncertainty of imputation, association test of the imputed genotypes was calculated with the SNPTEST v2 package[28].

**Gene expression analysis**. The mRNA expression data of 100 primary tumors and 29 cell lines were generated using Illumina Human 6 version 2 expression l bead chip. The mRNA expression data were previously published[29] and deposited in the Gene Expression Omnibus (GSE19274) database. 62 of the 100 neuroblastoma primary tumors have matched genotyping data.

**Sequencing data analysis**. Paired end exome sequencing data for 229 individual germline samples was obtained from NCI TARGET database (https://ocg.cancer.gov/programs/target/). Sample processing, exome capture and DNA sequencing methods are described in detail by Pugh et al.[30] and on the Target website (https://ocg.cancer.gov/programs/target-methods#3258). Fastq file sequences were aligned to the HG19/GRCh37 human genome sequence reference using BWA-mem[31]. Generated sam files were sorted, converted to bam format and duplicates removed using Samtools. Local realignment and variant calling including SNPs and short indels, was done using GATK IndelRealigner and UnifiedGenotyper respectively[32]. Whole-genome calls Complete Genomics masterVarBeta files containing variant calls for 143 neuroblastoma germline samples were obtained from the NCI TARGET database (https://ocg.cancer.gov/programs/target/). Sixty-five of the germline samples were also in the whole exome sequence sample set. The effect of normalized variants was annotated using dbNSFP[33], and population frequencies of variants were annotated using vcfanno[34] with 1000 Genomes databases. Here we report only MMP20 variants found in coding regions.

**Data availability**. Summary statistics of our genome-wide analysis can be downloaded from the public repository figshare (https://doi.org/10.6084/m9.figshare.4978145.v3)[35]. All other remaining data are available within the Article and Supplementary Files, or available from the authors upon request.

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

## Acknowledgements

We thank the patients and their families for their participation in this study. The study was supported by Institutional Development Funds from the CHOP, and by NIH grants U01HG006830 (H.H.) and R01CA124709 (J.M.M.). This work was also supported in part by US National Institutes of Health grants RC1MD004418 to the TARGET consortium, and CA98543 and U10 CA180899 to the COG (J.M.M.). In addition, this project was funded in part with Federal funds from the National Cancer Institute, National Institutes of Health, under Contract No. HHSN261200800001E (J.M.M.). The content of this publication does not necessarily reflect the views of policies of the Department of Health and Human Services, nor does mention of trade names, commercial products, or organizations imply endorsement by the US Government.

## Author contributions

H.H. and J.M.M. conceived and supervised the study, guided interpretation of results and helped preparation of the manuscript. X.C. and L.M. performed the association analysis with guidance from P.S. X.C., Y.Z., J.G., S.J.D., and E.F.A. performed gene expression and copy number analysis. M.A.D. and C.H. contributed to DNA sample collection; F.D.M. and H.Q. assisted with organizing the phenotype data; C.H., K.T., and C.K. performed

the SNP genotyping and QC. P.S., J.L., Z.W., Y.L., and Y.G. assisted in discussing and revising the manuscript. X.C. drafted the manuscript. All authors read and approved the final manuscript.

## Additional information

**Competing interests:** The authors declare no competing financial interests.

