## [Peer Review File · Nature Communications]

Reviewers' comments:

Reviewer #1 (Remarks to the Author): Expert in GWAS

Overall this is a very interesting study, eventually worthy of publication in Nat Comm. The paper nicely shows that common variants in the MMP20 locus (11q22.2) are associated exclusively with a subtype of neuroblastoma containing 11q deletions. The discussion of the other subtypes are also interesting and as a matter of presentation, the current format is a little misleading. The title and abstract don't contain any info about the other somatic aberrations investigated. Out of the blue on page 6 these additional results are discussed. Perhaps a more inclusive title and approach would work better. The other findings are also quite interesting.

Important queries

1. Figure 1 & Supplementary Fig4 – are they discovery only or meta-analyzed results? Assume discovery but please annotate carefully. It is a little difficult to follow the flow of the paper. The numbers are very hard to follow.

a. After QC there're 300 cases with matched blood/tumor (page 3- "We subsequently identified 113 11q-deletion cases of European-American ancestry from the 300 samples with available blood DNA genotyping data"). At page 4 the authors mention 260 MNA + 69 1p + 113 11q = 442 cases.

b. It is not clear why the authors only use 78 neuroblastoma cases without 11q deletion for the additional case/case analysis. Since most MNA and 11q are inversely related they could use the ~260 MNA (used in this study) as additional controls in the analysis of cases with vs without 11q del. This might increase significance more strongly indicating an independent role of this locus in 11q deletion cases

2. Why don't the MNA & 1p cases show an association for LMO1? MNA is associated with high risk and LMO1 is more prominent in high risk? (page 4 – Sup table 2).

3. For the discussion of the MMP20 locus: "As MMP20 is located within the commonly deleted 11q region in tumors" (page 4), isn't it true that the majority involve breakage at 11q23? Is the breakage always/most often around MMP20? Are these germline common variants involved in the somatic rearrangement of 11q or enriched due to LOH and tagging something regulatory in the MMP20 locus?

4. Since this superb group has done extensive profiling of DNA and RNA aberrations, why not also do eQTL analysis for MMP20 and other genes on 11q. Is MMP20 the only important gene on 11q? Or are the common variants tagging a regulatory region that influences another up or downstream gene?

5. The analysis is for population substructure, which in turns informs the stratification issue. As it is written it is confusing and suggests that the set of 192 becomes 44- a radical and disruptive change. This paragraph needs to be re-written carefully and the above point clarified. After reading it twice I was still confused.

Reviewer #2 (Remarks to the Author): Expert in GWAS

The authors have performed a GWAS and report the association between common genetic variation at 11q22.2 and 11q deletion in neuroblastoma. They implicate variation within MMP20 as the basis of the association since SNP associations map to this gene, an assertion supported by the observation that the risk allele defined by SNP rs10895322 is preferentially retained in tumors.

While this paper has potential interest there are issues:

1. It should be explicitly stated whether cases and controls were pruned to a common set of SNPs before imputation. If not this should be performed (see recent papers by Cordell and others).

2. The authors have clearly explored a number of subtypes for specific associations, and have not

solely sought to establish a relationship with 11q deletion. Hence the conventional threshold of 5.0×10^{-8} is not sufficient to address the multiple testing issue here. It should explicitly how many sub-groups they assessed and to what threshold was used.

3. Supplementary Table 3 should be within main body of the text. It shows the impact of the various risk loci on the subtypes. What about using output ASSET to show relationships (Am J Hum Genet; 90(5): 821)

4. What was the reason for failure to demonstrate associations for three previously identified risk loci?

5. Key and the major issue– there is copious amounts of data within the supplementary but almost all does not directly relate to understanding the biological basis of said association. While one acknowledges that the SNP association maps to MMP20, there is no data to support it being the functional basis of the association – the only reference to one published study which does not address functionality in the tumor. Is there sequencing data to exclude the possibility of association being a consequence of a rare variant? If the association is a consequence of an effect on differential gene expression, there should be functional data to support the assertion that MMP20 is the target (it is well known that looping interactions from the genomic region to which the SNP association emanates could link another gene). The functional studies appropriate to addressing this are well articulated by recent papers. Without some biology this paper provides little real insight into the biology of this cancer.

6. The main Figure in the text is less than informative. What about annotating it with some CHROMHM or such like tracks

Reviewer #3 (Remarks to the Author): Expert in neuroblastoma

The paper by et al. reports on the identification of common variants at 11q22.2 within MMP20 that associate with neuroblastoma cases harboring 11q deletion. To this end the authors conducted a GWAS in 113 European-American cases and 5,109 ancestry-matched controls. The association was subsequently replicated in 44 independent cases and 1,902 controls.

Based on the observation that the lead SNP rs10895322 was also the top SNP reported in the GWAS of neovascular lesion size in AMD and the risk allele of rs10895322 in this study being the same allele that promotes neovascular lesion size in the GWAS of AMD, the authors propose that the risk attributed to rs10895322 in neuroblastoma could be mediated through a comparable angiogenesis process regulated by MMP20.

This paper adds further to previous GWAS studies in neuroblastoma patients performed by this team and which have lead to several important novel insights. Here, SNP rs10895322 was identified to be associated to NB cases harbouring 11q deletions. This is genetically still an enigmatic subgroup of patients with poor prognosis. Typically large 11q deletions mark this group often with 17q gain and additional focal gains and losses but particular driver or suppressors still remain to be determined. The data presented here suggested that MMP20 could play a role in the biology of this subgroup.

Finally, the authors also describe their findings on rare amplicons and chromothripsis in the cohort investigated in this study.

Comments:

1. There is little information regarding the potential functional contribution of this SNP to (increased risk for) tumor formation apart from data available from the literature which putatively connect MMP20 to angiogenesis. Can the authors provide more information on the putative effect of the SNP on expression levels or function.

2. What is known on MMP20 in relation to effect of mRNA levels in patient tumor samples, in particular

are distinct differences noted between 11q deleted versus normal tumors?

3. The last section on the rare amplicons detected in this new tumor cohort is of some interest but somewhat disconnected from the key message of this paper in my view.

Reviewers' comments:

Reviewer #1 (Remarks to the Author): Expert in GWAS

Overall this is a very interesting study, eventually worthy of publication in Nat Comm. The paper nicely shows that common variants in the MMP20 locus (11q22.2) are associated exclusively with a subtype of neuroblastoma containing 11q deletions. The discussion of the other subtypes are also interesting and as a matter of presentation, the current format is a little misleading. The title and abstract don't contain any info about the other somatic aberrations investigated. Out of the blue on page 6 these additional results are discussed. Perhaps a more inclusive title and approach would work better. The other findings are also quite interesting.

Important queries

1. Figure 1 & Supplementary Fig4 – are they discovery only or meta-analyzed results? Assume discovery but please annotate carefully. It is a little difficult to follow the flow of the paper. The numbers are very hard to follow.

Response: Associations and markers showed in Fig 1 and Supplementary Fig 4 are the meta-analyzed results. We have clarified this in the description of Figure 1 and Supplementary Figure 4 in the revised manuscript.

a. After QC there're 300 cases with matched blood/tumor (page 3- “We subsequently identified 113 11q-deletion cases of European-American ancestry from the 300 samples with available blood DNA genotyping data”). At page 4 the authors mention $260 \text{ MNA} + 69 \text{ 1p} + 113 \text{ 11q} = 442$ cases.

Response: In the 300 cases passing QC, 125, 171 and 101 present 1p deletion, 11q deletion and MNA, respectively ($125 \text{ 1p} + 171 \text{ 11q} + 101 \text{ MNA} = 377$). The number is bigger than 300, since 1p deletion co-occurs with MNA or 11q deletion in some neuroblastoma cases. Among the 300 cases, we further identified 78 1p-deletion, 113 11q-deletion, and 56 MNA cases of European ancestry for the association analysis. For the MNA GWAS, we also included additional 299 genotyped MNA cases, which were identified by fluorescence in situ hybridization as previously described (PMID: 11420745). 204 of the 299 MNA cases are European ancestry. Finally, 260 ($56 + 204 = 260$) MNA samples of European ancestry were included for the association analysis. A detailed

description of the cases and subtypes has been included in the Methods (Page 9-10. Line 234-245).

b. It is not clear why the authors only use 78 neuroblastoma cases without 11q deletion for the additional case/case analysis. Since most MNA and 11q are inversely related they could use the ~260 MNA (used in this study) as additional controls in the analysis of cases with vs without 11q del. This might increase significance more strongly indicating an independent role of this locus in 11q deletion cases

Response: There are only 78 undeleted 11q neuroblastoma cases of the European ancestry identified from the 300 cases with matched blood/tumor DNAs. However, we do have 204 additional MNA cases of European ancestry identified by fluorescence in situ hybridization as described previously (PMID: 11420745). We reanalyzed the 113 11q-deletion cases and 282 controls (78 11q undeleted and 204 additional MNA neuroblastomas) as suggested by the reviewer. Indeed, a more significant association was detected ($P = 5.50 \times 10^{-5}$). We have updated the manuscript and Supplementary Table 5 accordingly (Page 5. Line 116-118).

2. Why don't the MNA & 1p cases show an association for LMO1? MNA is associated with high risk and LMO1 is more prominent in high risk? (page 4 – Sup table 2).

Response: The *LMO1* loci was first reported by Wang et al (PMID: 21124317). The authors analyzed the risk allele of *LMO1* with clinical variables such as risk levels and MNA statuses. They found the *LMO1* risk allele is enriched in the high-risk neuroblastoma cases, but not in the MNA cases. Similar results were found in this study, suggesting that *LMO1* may be more prominent in the subgroup of high-risk neuroblastoma, which do not present MNA. For the 1p-deletion GWAS, the number of cases is very small (69 cases). Lack of association with *LMO1* in 1p-del GWAS cases may be due to lack of power.

3. For the discussion of the MMP20 locus: “As MMP20 is located within the commonly deleted 11q region in tumors” (page 4), isn't it true that the majority involve breakage at 11q23? Is the breakage always/most often around MMP20? Are these germline common variants involved in the somatic rearrangement of 11q or enriched due to LOH and tagging something regulatory in the MMP20 locus?

Response: The breakage at 11q23 was initially reported by Guo et al (PMID: 10490829). A follow up study by Maris et al found that the commonly deleted region is 11q14-23 (PMID: 11464895). Here, our high-resolution SNP array showed that the deleted region is even larger, and the breakpoint is at 11q13. So the *MMP20* locus is located within the commonly deleted region of 11q. Since the location of *MMP20*, 11q22.2, is distant from the breakage at 11q13, there is no evidence suggest that the germline common variants may be involved in the somatic rearrangement of 11q. In addition, we did not find evidence suggest that the reported common variants is enriched due to LOH. However, those variants may be involved in the regulation of genes at the *MMP20* locus. For example, CHROMHMM tracks of the updated Figure1 suggest the presented variants are located in a regulatory region of multiple cell lines.

4. Since this superb group has done extensive profiling of DNA and RNA aberrations, why not also do eQTL analysis for *MMP20* and other genes on 11q. Is *MMP20* the only important gene on 11q? Or are the common variants tagging a regulatory region that influences another up or downstream gene?

Response: We analyzed mRNA expression from 100 primary neuroblastoma tumors (GSE19274). 62 of them have matched SNP array data and were included in the current study. Among the 62 neuroblastoma samples, 34 present 11q deletion. eQTL analysis show that rs10895322 genotypes are significantly associated with *MMP20* expression in 11q-deletion subtypes ($P=7.7 \times 10^{-5}$). However, there is no significant association between rs10895322 genotypes and the expression level of other genes near *MMP20* in the 11q-deletion subtype. In addition, no association was detected between rs10895322 genotypes and *MMP20* expression in 11q undeleted neuroblastoma samples ($P=0.94$). We added the description of the eQTL analysis at (Page 7, Line 165-174), and made a new supplementary figure 6 to show the eQTL results.

5. The analysis is for population substructure, which in turns informs the stratification issue. As it is written it is confusing and suggests that the set of 192 becomes 44- a radical and disruptive change. This paragraph needs to be re-written carefully and the above point clarified. After reading it twice I was still confused.

Response: For the replication study, we first genotyped the matched blood and tumor DNA samples from 192 neuroblastoma cases. Among them, only 80 showed 11q deletion. Since the 80 11q-del cases are recruited from the general population within the Philadelphia area, African Americans, Hispanic and Latino Americans, and European Americans were included. We further performed population stratification analysis by Eigenstrat, and identified 44 patients are European descendants. The 44 neuroblastoma cases were used in the replication study. We have re-written the description of this part (Page 6, Line 133-137).

Reviewer #2 (Remarks to the Author): Expert in GWAS

The authors have performed a GWAS and report the association between common genetic variation at 11q22.2 and 11q deletion in neuroblastoma. They implicate variation within MMP20 as the basis of the association since SNP associations map to this gene, an assertion supported by the observation that the risk allele defined by SNP rs10895322 is preferentially retained in tumors.

While this paper has potential interest there are issues:

1. It should be explicitly stated whether cases and controls were pruned to a common set of SNPs before imputation. If not this should be performed (see recent papers by Cordell and others).

Response: Yes. The GWAS was pruned to a common set of SNPs between cases and controls before imputation. We have clarified this in the method (Page 11, Line 263).

2. The authors have clearly explored a number of subtypes for specific associations, and have not solely sought to establish a relationship with 11q deletion. Hence the conventional threshold of 5.0×10^{-8} is not sufficient to address the multiple testing issue here. It should explicitly how many subgroups they assessed and to what threshold was used.

Response: We evaluated three subtypes of neuroblastoma including *MYCN* amplification, 11q deletion, and 1p deletion. The conventional *P* value threshold of 5.0×10^{-8} was used in this study. We agree with the reviewer that the conventional threshold may not be sufficient to address the multiple testing issue. Considering three subtypes were investigated, we modified the

conventional threshold by dividing it by three (1.7×10^{-8}). The top SNP rs10895322 is still significant ($P = 2.62 \times 10^{-9}$). We have modified the description of this part accordingly (Page 4, Line 90-93).

3. Supplementary Table 3 should be within main body of the text. It shows the impact of the various risk loci on the subtypes. What about using output ASSET to show relationships (Am J Hum Genet; 90(5): 821)

Response: We performed the association analysis using the subset-based approach incorporated in ASSET (Am J Hum Genet; 90(5): 821) as suggested by the reviewer. ASSET is designed for a single case-control study in which cases can consist of distinct disease subtypes. This permits both case-control and case-case comparisons (among subsets of disease subtypes) to detect the strongest association signals. In this study, 11q deletion and *MYCN* amplification can be considered as two distinct subtypes of neuroblastoma, since they are negatively correlated. However, 1p deletion is not an independent subtype, since 1p deletion co-occurs with 11q deletion or *MYCN* amplification. We therefore ran ASSET on the cases presenting 11q-deletion or *MYCN* amplification. ASSET results confirm that 11q22.2 is associated with 11q-deletion subtype. We also made a new table (Table 2) and added it into the main body of text (Page 5-6, Line 120-131).

4. What was the reason for failure to demonstrate associations for three previously identified risk loci?

Response: First, most of cases in this study are high-risk neuroblastoma. Thus, the genes *BARD1*, *CASC15* and *LMO1*, which are associated with high-risk neuroblastoma (PMID: 21436895) were uncovered. However, *HSD17B12* is mainly associated with low-risk neuroblastoma (PMID: 21436895). *HACE* and *LIN28B* are associated with all risk stages of neuroblastoma (PMID: 22941191). Failure to demonstrate the three loci may be due to their limited impact in the high-risk neuroblastoma. The second reason may be due to lack of power. As reported in the previous GWAS (PMID: 22941191), the signals of *BARD1* (4.14×10^{-14}), *CASC15* (7.82×10^{-16}) and *LMO1* (1.26×10^{-13}) are considerably stronger than those of *HACE1* (1.8×10^{-8}), *LIN28B* (1.8×10^{-7}) and *HSD17B12* (4.89×10^{-8}).

5. Key and the major issue– there is copious amounts of data within the supplementary but almost all does not directly relate to understanding the biological basis of said association. While one acknowledges that the SNP

association maps to *MMP20*, there is no data to support it being the functional basis of the association – the only reference to one published study which does not address functionality in the tumor. Is there sequencing data to exclude the possibility of association being a consequence of a rare variant? If the association is a consequence of an effect on differential gene expression, there should be functional data to support the assertion that *MMP20* is the target (it is well known that looping interactions from the genomic region to which the SNP association emanates could link another gene). The functional studies appropriate to addressing this are well articulated by recent papers. Without some biology this paper provides little real insight into the biology of this cancer.

Response: To investigate the possibility that the association being a consequence of a rare variant, we analyzed exome sequencing data of 229 neuroblastoma samples and whole genome sequencing data of 143 neuroblastoma samples from the TARGET database (<https://ocg.cancer.gov/programs/target/>). We checked low frequency variants (<0.01 in 1000 Genomes Database) detected in the *MMP20* coding region. The variants were summarized in Supplementary Table 7. Only one pathogenic variant was detected, which is associated with amelogenesis imperfecta. In addition, the frequency of rare variants presents in neuroblastoma cases is not significantly different in the 1000 Genome Database, indicating that the association is unlikely a consequence of a rare variant. We added the description of the sequencing results at Page 6-7, Line 153-163.

To investigate the effect of identified variants on gene expression, we analyzed the mRNA expression data of 100 primary neuroblastoma tumors (GSE19274). 62 of them have matched SNP array data and were included in the current study. Among the 62 neuroblastoma samples, 34 present 11q deletion. eQTL analysis show that rs10895322 is significantly associated with *MMP20* expression in 11q-deletion subtypes ($P=7.7 \times 10^{-5}$). However, there is no significant association between rs10895322 genotypes and the expression level of other genes near *MMP20* in the 11q-deletion subtype. In addition, no association was observed between rs10895322 and *MMP20* expression in 11q undeleted neuroblastoma samples ($P=0.94$). We added the description of the eQTL analysis at Page 7, Line 165-174, and made a new supplementary figure 6 to show the eQTL results.

6. The main Figure in the text is less than informative. What about

annotating it with some CHROMHM or such like tracks

Response: We have modified Figure 1 in the main text with adding CHROMHMM tracks (Celline: HepG2, HSMM, K562, GM12878), wgEncodeRegTfbsClusteredV3 and tfbsConsSites as suggested by the reviewer.

Reviewer #3 (Remarks to the Author): Expert in neuroblastoma

The paper by et al. reports on the identification of common variants at 11q22.2 within MMP20 that associate with neuroblastoma cases harboring 11q deletion. To this end the authors conducted a GWAS in 113 European-American cases and 5,109 ancestry-matched controls. The association was subsequently replicated in 44 independent cases and 1,902 controls. Based on the observation that the lead SNP rs10895322 was also the top SNP reported in the GWAS of neovascular lesion size in AMD and the risk allele of rs10895322 in this study being the same allele that promotes neovascular lesion size in the GWAS of AMD, the authors propose that the risk attributed to rs10895322 in neuroblastoma could be mediated through a comparable angiogenesis process regulated by MMP20.

This paper adds further to previous GWAS studies in neuroblastoma patients performed by this team and which have lead to several important novel insights. Here, SNP rs10895322 was identified to be associated to NB cases harbouring 11q deletions. This is genetically still an enigmatic subgroup of patients with poor prognosis. Typically large 11q deletions mark this group often with 17q gain and additional focal gains and losses but particular driver or suppressors still remain to be determined. The data presented here suggested that MMP20 could play a role in the biology of this subgroup.

Finally, the authors also describe their findings on rare amplicons and chromothripsis in the cohort investigated in this study.

Comments:

1. There is little information regarding the potential functional contribution of this SNP to (increased risk for) tumor formation apart from data available from the literature which putatively connect MMP20 to angiogenesis. Can the authors provide more information on the putative effect of the SNP on

expression levels or function.

Response: We analyzed the mRNA expression data from 100 primary neuroblastoma tumors (GSE19274). 62 of them have matched SNP array data and were included in the current study. Among the 62 neuroblastoma samples, 34 present 11q deletion. eQTL analysis shows that rs10895322 is significantly associated with *MMP20* expression in 11q-deletion subtypes ($P=7.7 \times 10^{-5}$). However, there is no association between rs10895322 genotypes and the expression level of other genes near *MMP20* in the 11q-deletion subtype. In addition, no association was observed between rs10895322 and *MMP20* expression in 11q undeleted neuroblastoma samples ($P=0.94$). We added the description of the eQTL analysis at (Page 7, Line 165-174), and made a new supplementary figure 6 to show the eQTL results.

2. What is known on *MMP20* in relation to effect of mRNA levels in patient tumor samples, in particular are distinct differences noted between 11q deleted versus normal tumors?

Response: We analyzed the *MMP20* expression in 34 11q-deletion vs 28 11q undeleted cases. No significant expression difference was detected between 11q-deletion and 11q-undeleted neuroblastoma samples ($P=0.26$).

3. The last section on the rare amplicons detected in this new tumor cohort is of some interest but somewhat disconnected from the key message of this paper in my view.

Response: We revised this section and integrated it into Page 3-4, Line 67-78. Details of the rare amplicons and chromothripsis have been moved to the supplementary notes.

REVIEWERS' COMMENTS:

Reviewer #1 (Remarks to the Author):

the authors have adeptly answered both this reviewer's and the others' suggestions.
No further requests at this time.

Reviewer #2 (Remarks to the Author):

The authors have addressed my concerns satisfactorily